# ProxylessNAS: Direct Neural Architecture Search on Target Task and Hardware

**Han Cai, Ligeng Zhu, Song Han**
Massachusetts Institute of Technology
{hancai, ligeng, songhan}@mit.edu

## Abstract

Neural architecture search (NAS) has a great impact by automatically designing effective neural network architectures. However, the prohibitive computational demand of conventional NAS algorithms (e.g. $10^4$ GPU hours) makes it difficult to *directly* search the architectures on large-scale tasks (e.g. ImageNet). Differentiable NAS can reduce the cost of GPU hours via a continuous representation of network architecture but suffers from the high GPU memory consumption issue (grow linearly w.r.t. candidate set size). As a result, they need to utilize *proxy* tasks, such as training on a smaller dataset, or learning with only a few blocks, or training just for a few epochs. These architectures optimized on proxy tasks are not guaranteed to be optimal on the target task. In this paper, we present *ProxylessNAS* that can *directly* learn the architectures for large-scale target tasks and target hardware platforms. We address the high memory consumption issue of differentiable NAS and reduce the computational cost (GPU hours and GPU memory) to the same level of regular training while still allowing a large candidate set. Experiments on CIFAR-10 and ImageNet demonstrate the effectiveness of directness and specialization. On CIFAR-10, our model achieves 2.08% test error with only 5.7M parameters, better than the previous state-of-the-art architecture AmoebaNet-B, while using $6\times$ fewer parameters. On ImageNet, our model achieves 3.1% better top-1 accuracy than MobileNetV2, while being $1.2\times$ faster with measured GPU latency. We also apply ProxylessNAS to specialize neural architectures for hardware with direct hardware metrics (e.g. latency) and provide insights for efficient CNN architecture design.[1]

## 1 Introduction

Neural architecture search (NAS) has demonstrated much success in automating neural network architecture design for various deep learning tasks, such as image recognition (Zoph et al., 2018; Cai et al., 2018a; Liu et al., 2018a; Zhong et al., 2018) and language modeling (Zoph & Le, 2017). Despite the remarkable results, conventional NAS algorithms are prohibitively computation-intensive, requiring to train thousands of models on the target task in a single experiment. Therefore, directly applying NAS to a large-scale task (e.g. ImageNet) is computationally expensive or impossible, which makes it difficult for making practical industry impact. As a trade-off, Zoph et al. (2018) propose to search for building blocks on proxy tasks, such as training for fewer epochs, starting with a smaller dataset (e.g. CIFAR-10), or learning with fewer blocks. Then top-performing blocks are stacked and transferred to the large-scale target task. This paradigm has been widely adopted in subsequent NAS algorithms (Liu et al., 2018a;b; Real et al., 2018; Cai et al., 2018b; Liu et al., 2018c; Tan et al., 2018; Luo et al., 2018).

However, these blocks optimized on proxy tasks are not guaranteed to be optimal on the target task, especially when taking hardware metrics such as latency into consideration. More importantly, to enable transferability, such methods need to search for only a few architectural motifs and then repeatedly stack the same pattern, which restricts the block diversity and thereby harms performance.

In this work, we propose a simple and effective solution to the aforementioned limitations, called *ProxylessNAS*, which directly learns the architectures on the target task and hardware instead of with

---

[1] Pretrained models and evaluation code are released at https://github.com/MIT-HAN-LAB/ProxylessNAS.

(1) Previous proxy-based approach

(2) Our proxy-less approach

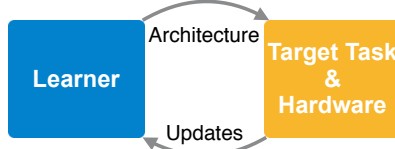

Figure 1: ProxylessNAS directly optimizes neural network architectures on target task and hardware. Benefiting from the directness and specialization, ProxylessNAS can achieve remarkably better results than previous proxy-based approaches. On ImageNet, with only 200 GPU hours (200 × fewer than MnasNet (Tan et al., 2018)), our searched CNN model for mobile achieves the same level of top-1 accuracy as MobileNetV2 1.4 while being 1.8× faster.

proxy (Figure 1). We also remove the restriction of repeating blocks in previous NAS works (Zoph et al., 2018; Liu et al., 2018c) and allow all of the blocks to be learned and specified. To achieve this, we reduce the computational cost (GPU hours and GPU memory) of architecture search to the same level of regular training in the following ways.

GPU hour-wise, inspired by recent works (Liu et al., 2018c; Bender et al., 2018), we formulate NAS as a path-level pruning process. Specifically, we directly train an over-parameterized network that contains all candidate paths (Figure 2). During training, we explicitly introduce architecture parameters to learn which paths are redundant, while these redundant paths are pruned at the end of training to get a compact optimized architecture. In this way, we only need to train a single network without any meta-controller (or hypernetwork) during architecture search.

However, naively including all the candidate paths leads to GPU memory explosion (Liu et al., 2018c; Bender et al., 2018), as the memory consumption grows linearly w.r.t. the number of choices. Thus, GPU memory-wise, we binarize the architecture parameters (1 or 0) and force only one path to be active at run-time, which reduces the required memory to the same level of training a compact model. We propose a gradient-based approach to train these binarized parameters based on BinaryConnect (Courbariaux et al., 2015). Furthermore, to handle non-differentiable hardware objectives (using latency as an example) for learning specialized network architectures on target hardware, we model network latency as a continuous function and optimize it as regularization loss. Additionally, we also present a REINFORCE-based (Williams, 1992) algorithm as an alternative strategy to handle hardware metrics.

In our experiments on CIFAR-10 and ImageNet, benefiting from the directness and specialization, our method can achieve strong empirical results. On CIFAR-10, our model reaches 2.08% test error with only 5.7M parameters. On ImageNet, our model achieves 75.1% top-1 accuracy which is 3.1% higher than MobileNetV2 (Sandler et al., 2018) while being 1.2× faster. Our contributions can be summarized as follows:

- ProxylessNAS is the first NAS algorithm that directly learns architectures on the large-scale dataset (e.g. ImageNet) without any proxy while still allowing a large candidate set and removing the restriction of repeating blocks. It effectively enlarged the search space and achieved better performance.

- We provide a new path-level pruning perspective for NAS, showing a close connection between NAS and model compression (Han et al., 2016). We save memory consumption by one order of magnitude by using path-level binarization.

- We propose a novel gradient-based approach (latency regularization loss) for handling hardware objectives (e.g. latency). Given different hardware platforms: CPU/GPU/Mobile, ProxylessNAS enables hardware-aware neural network specialization that's exactly optimized for the target hardware. To our best knowledge, it is the first work to study specialized *neural network architectures* for different *hardware architectures*.

- Extensive experiments showed the advantage of the directness property and the specialization property of ProxylessNAS. It achieved state-of-the-art accuracy performances on CIFAR-10 and ImageNet under latency constraints on different hardware platforms (GPU, CPU and mobile phone). We also analyze the insights of efficient CNN models specialized for different hardware platforms and raise the awareness that specialized neural network architecture is needed on different hardware architectures for efficient inference.

## 2  RELATED WORK

The use of machine learning techniques, such as reinforcement learning or neuro-evolution, to re-place human experts in designing neural network architectures, usually referred to as neural archi-tecture search, has drawn an increasing interest (Zoph & Le, 2017; Liu et al., 2018a;b;c; Cai et al., 2018a;b; Pham et al., 2018; Brock et al., 2018; Bender et al., 2018; Elsken et al., 2017; 2018b; Ka-math et al., 2018). In NAS, architecture search is typically considered as a meta-learning process, and a meta-controller (e.g. a recurrent neural network (RNN)), is introduced to explore a given architecture space with training a network in the inner loop to get an evaluation for guiding explo-ration. Consequently, such methods are computationally expensive to run, especially on large-scale tasks, e.g. ImageNet.

Some recent works (Brock et al., 2018; Pham et al., 2018) try to improve the efficiency of this meta-learning process by reducing the cost of getting an evaluation. In Brock et al. (2018), a hy-pernetwork is utilized to generate weights for each sampled network and hence can evaluate the architecture without training it. Similarly, Pham et al. (2018) propose to share weights among all sampled networks under the standard NAS framework (Zoph & Le, 2017). These methods speed up architecture search by orders of magnitude, however, they require a hypernetwork or an RNN controller and mainly focus on small-scale tasks (e.g. CIFAR) rather than large-scale tasks (e.g. ImageNet).

Our work is most closely related to One-Shot (Bender et al., 2018) and DARTS (Liu et al., 2018c), both of which get rid of the meta-controller (or hypernetwork) by modeling NAS as a single training process of an over-parameterized network that comprises all candidate paths. Specifically, One-Shot trains the over-parameterized network with DropPath (Zoph et al., 2018) that drops out each path with some fixed probability. Then they use the pre-trained over-parameterized network to evaluate architectures, which are sampled by randomly zeroing out paths. DARTS additionally introduces a real-valued architecture parameter for each path and jointly train weight parameters and architecture parameters via standard gradient descent. However, they suffer from the large GPU memory consumption issue and hence still need to utilize proxy tasks. In this work, we address the large memory issue in these two methods through path binarization.

Another relevant topic is network pruning (Han et al., 2016) that aim to improve the efficiency of neural networks by removing insignificant neurons (Han et al., 2015) or channels (Liu et al., 2017). Similar to these works, we start with an over-parameterized network and then prune the redundant parts to derive the optimized architecture. The distinction is that they focus on layer-level pruning that only modifies the filter (or units) number of a layer but can not change the topology of the network, while we focus on learning effective network architectures through *path-level pruning*. We also allow both pruning and growing the number of layers.

## 3  METHOD

We first describe the construction of the over-parameterized network with all candidate paths, then introduce how we leverage binarized architecture parameters to reduce the memory consumption of training the over-parameterized network to the same level as regular training. We propose a gradient-based algorithm to train these binarized architecture parameters. Finally, we present two techniques to handle non-differentiable objectives (e.g. latency) for specializing neural networks on target hardware.

### 3.1  CONSTRUCTION OF OVER-PARAMETERIZED NETWORK

Denote a neural network as $\mathcal{N}(e, \cdots, e_n)$ where $e_i$ represents a certain edge in the directed acyclic graph (DAG). Let $\mathcal{O} = \{o_i\}$ be the set of $N$ candidate primitive operations (e.g. convolution, pool-ing, identity, zero, etc). To construct the over-parameterized network that includes any architecture in the search space, instead of setting each edge to be a definite primitive operation, we set each edge to be a mixed operation that has $N$ parallel paths (Figure 2), denoted as $m_{\mathcal{O}}$. As such, the over-parameterized network can be expressed as $\mathcal{N}(e = m_{\mathcal{O}}^1, \cdots, e_n = m_{\mathcal{O}}^n)$.

Given input $x$, the output of a mixed operation $m_{\mathcal{O}}$ is defined based on the outputs of its $N$ paths. In One-Shot, $m_{\mathcal{O}}(x)$ is the sum of $\{o_i(x)\}$, while in DARTS, $m_{\mathcal{O}}(x)$ is weighted sum of $\{o_i(x)\}$ where

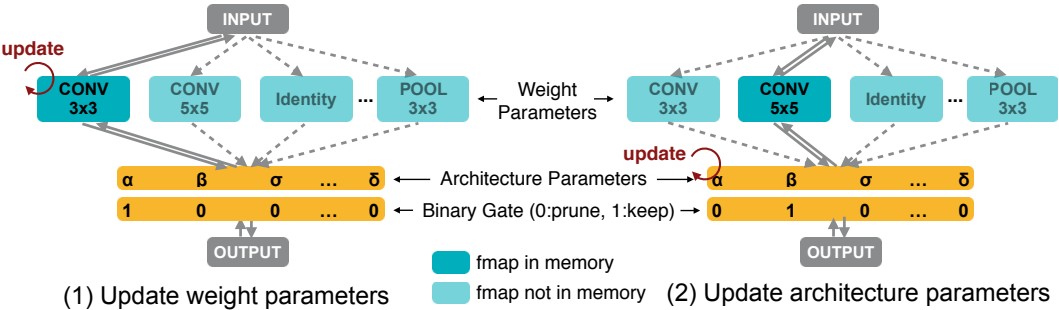

Figure 2: Learning both weight parameters and binarized architecture parameters.

the weights are calculated by applying softmax to $N$ real-valued architecture parameters $\{\alpha_i\}$:

$$m_{\mathcal{O}}^{\text{One-Shot}}(x) = \sum_{i=1}^{N} o_i(x), \qquad m_{\mathcal{O}}^{\text{DARTS}}(x) = \sum_{i=1}^{N} p_i o_i(x) = \sum_{i=1}^{N} \frac{\exp(\alpha_i)}{\sum_j \exp(\alpha_j)} o_i(x). \qquad (1)$$

As shown in Eq. (1), the output feature maps of all N paths are calculated and stored in the memory, while training a compact model only involves one path. Therefore, One-Shot and DARTS roughly need $N$ times GPU memory and GPU hours compared to training a compact model. On large-scale dataset, this can easily exceed the memory limits of hardware with large design space. In the following section, we solve this memory issue based on the idea of path binarization.

## 3.2 LEARNING BINARIZED PATH

To reduce memory footprint, we keep only one path when training the over-parameterized network. Unlike Courbariaux et al. (2015) which binarize individual weights, we binarize entire paths. We introduce $N$ real-valued architecture parameters $\{\alpha_i\}$ and then transforms the real-valued path weights to binary gates:

$$g = \text{binarize}(p_1, \cdots, p_N) = \begin{cases} [1, 0, \cdots, 0] & \text{with probability } p_1, \\ \quad \cdots \\ [0, 0, \cdots, 1] & \text{with probability } p_N. \end{cases} \qquad (2)$$

Based on the binary gates $g$, the output of the mixed operation is given as:

$$m_{\mathcal{O}}^{\text{Binary}}(x) = \sum_{i=1}^{N} g_i o_i(x) = \begin{cases} o_1(x) & \text{with probability } p_1 \\ \quad \cdots \\ o_N(x) & \text{with probability } p_N. \end{cases} \qquad (3)$$

As illustrated in Eq. (3) and Figure 2, by using the binary gates rather than real-valued path weights (Liu et al., 2018c), only one path of activation is active in memory at run-time and the memory requirement of training the over-parameterized network is thus reduced to the same level of training a compact model. That's more than an order of magnitude memory saving.

### 3.2.1 TRAINING BINARIZED ARCHITECTURE PARAMETERS

Figure 2 illustrates the training procedure of the weight parameters and binarized architecture parameters in the over-parameterized network. When training weight parameters, we first freeze the architecture parameters and stochastically sample binary gates according to Eq. (2) for each batch of input data. Then the weight parameters of active paths are updated via standard gradient descent on the training set (Figure 2 left). When training architecture parameters, the weight parameters are frozen, then we reset the binary gates and update the architecture parameters on the validation set (Figure 2 right). These two update steps are performed in an alternative manner. Once the training of architecture parameters is finished, we can then derive the compact architecture by pruning redundant paths. In this work, we simply choose the path with the highest path weight.

Unlike weight parameters, the architecture parameters are not directly involved in the computation graph and thereby cannot be updated using the standard gradient descent. In this section, we introduce a gradient-based approach to learn the architecture parameters.

In BinaryConnect (Courbariaux et al., 2015), the real-valued weight is updated using the gradient w.r.t. its corresponding binary gate. In our case, analogously, the gradient w.r.t. architecture parameters can be approximately estimated using $\partial L / \partial g_i$ in replace of $\partial L / \partial p_i$:

$$\frac{\partial L}{\partial \alpha_i} = \sum_{j=1}^{N} \frac{\partial L}{\partial p_j} \frac{\partial p_j}{\partial \alpha_i} \approx \sum_{j=1}^{N} \frac{\partial L}{\partial g_j} \frac{\partial p_j}{\partial \alpha_i} = \sum_{j=1}^{N} \frac{\partial L}{\partial g_j} \frac{\partial \left( \frac{\exp(\alpha_j)}{\sum_k \exp(\alpha_k)} \right)}{\partial \alpha_i} = \sum_{j=1}^{N} \frac{\partial L}{\partial g_j} p_j (\delta_{ij} - p_i), \quad (4)$$

where $\delta_{ij} = 1$ if $i = j$ and $\delta_{ij} = 0$ if $i \neq j$. Since the binary gates $g$ are involved in the computation graph, as shown in Eq. (3), $\partial L / \partial g_j$ can be calculated through backpropagation. However, computing $\partial L / \partial g_j$ requires to calculate and store $o_j(x)$. Therefore, directly using Eq. (4) to update the architecture parameters would also require roughly $N$ times GPU memory compared to training a compact model.

To address this issue, we consider factorizing the task of choosing one path out of N candidates into multiple binary selection tasks. The intuition is that if a path is the best choice at a particular position, it should be the better choice when solely compared to any other path.[2]

Following this idea, within an update step of the architecture parameters, we first sample two paths according to the multinomial distribution $(p_1, \cdots, p_N)$ and mask all the other paths as if they do not exist. As such the number of candidates temporarily decrease from $N$ to 2, while the path weights $\{p_i\}$ and binary gates $\{g_i\}$ are reset accordingly. Then we update the architecture parameters of these two sampled paths using the gradients calculated via Eq. (4). Finally, as path weights are computed by applying softmax to the architecture parameters, we need to rescale the value of these two updated architecture parameters by multiplying a ratio to keep the path weights of unsampled paths unchanged. As such, in each update step, one of the sampled paths is enhanced (path weight increases) and the other sampled path is attenuated (path weight decreases) while all other paths keep unchanged. In this way, regardless of the value of $N$, only two paths are involved in each update step of the architecture parameters, and thereby the memory requirement is reduced to the same level of training a compact model.

### 3.3 HANDLING NON-DIFFERENTIABLE HARDWARE METRICS

Besides accuracy, latency (not FLOPs) is another very important objective when designing efficient neural network architectures for hardware. Unfortunately, unlike accuracy that can be optimized using the gradient of the loss function, latency is non-differentiable. In this section, we present two algorithms to handle the non-differentiable objectives.

#### 3.3.1 MAKING LATENCY DIFFERENTIABLE

To make latency differentiable, we model the latency of a network as a continuous function of the neural network dimensions [3]. Consider a mixed operation with a candidate set $\{o_j\}$ and each $o_j$ is associated with a path weight $p_j$ which represents the probability of choosing $o_j$. As such, we have the expected latency of a mixed operation (i.e. a learnable block) as:

$$\mathbb{E}[\text{latency}_i] = \sum_j p_j^i \times F(o_j^i), \quad (5)$$

where $\mathbb{E}[\text{latency}_i]$ is the expected latency of the $i^{th}$ learnable block, $F(\cdot)$ denotes the latency prediction model and $F(o_j^i)$ is the predicted latency of $o_j^i$. The gradient of $\mathbb{E}[\text{latency}_i]$ w.r.t. architecture parameters can thereby be given as: $\partial \mathbb{E}[\text{latency}_i] / \partial p_j^i = F(o_j^i)$.

For the whole network with a sequence of mixed operations (Figure 3 left), since these operations are executed sequentially during inference, the expected latency of the network can be expressed with the sum of these mixed operations' expected latencies:

$$\mathbb{E}[\text{latency}] = \sum_i \mathbb{E}[\text{latency}_i], \quad (6)$$

---

[2]In Appendix D, we provide another solution to this issue that does not require the approximation.
[3]Details of the latency prediction model are provided in Appendix B.

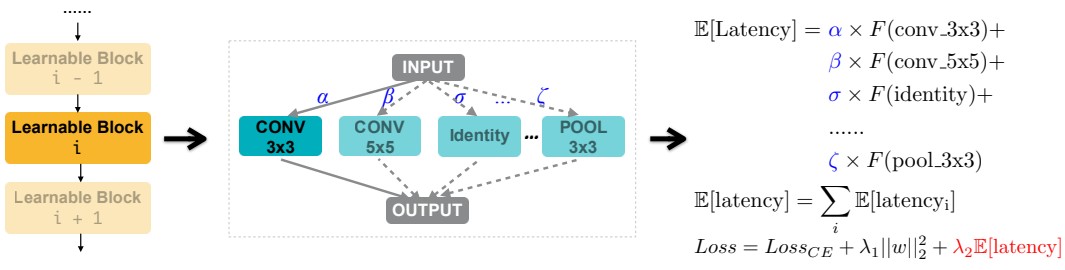

Figure 3: Making latency differentiable by introducing latency regularization loss.

We incorporate the expected latency of the network into the normal loss function by multiplying a scaling factor $\lambda_2(> 0)$ which controls the trade-off between accuracy and latency. The final loss function is given as (also shown in Figure 3 right)

$$Loss = Loss_{CE} + \lambda_1||w||_2^2 + \lambda_2\mathbb{E}[\text{latency}], \tag{7}$$

where $Loss_{CE}$ denotes the cross-entropy loss and $\lambda_1||w||_2^2$ is the weight decay term.

### 3.3.2 REINFORCE-BASED APPROACH

As an alternative to BinaryConnect, we can utilize REINFORCE to train binarized weights as well. Consider a network that has binarized parameters $\alpha$, the goal of updating binarized parameters is to find the optimal binary gates $g$ that maximizes a certain reward, denoted as $R(\cdot)$. Here we assume the network only has one mixed operation for ease of illustration. Therefore, according to REINFORCE (Williams, 1992), we have the following updates for binarized parameters:

$$J(\alpha) = \mathbb{E}_{g\sim\alpha}[R(\mathcal{N}_g)] = \sum_i p_i R(\mathcal{N}(e = o_i)),$$

$$\nabla_\alpha J(\alpha) = \sum_i R(\mathcal{N}(e = o_i))\nabla_\alpha p_i = \sum_i R(\mathcal{N}(e = o_i))p_i\nabla_\alpha \log(p_i),$$

$$= \mathbb{E}_{g\sim\alpha}[R(\mathcal{N}_g)\nabla_\alpha \log(p(g))] \approx \frac{1}{M}\sum_{i=1}^M R(\mathcal{N}_{g^i})\nabla_\alpha \log(p(g^i)), \tag{8}$$

where $g^i$ denotes the $i^{th}$ sampled binary gates, $p(g^i)$ denotes the probability of sampling $g^i$ according to Eq. (2) and $\mathcal{N}_{g^i}$ is the compact network according to the binary gates $g^i$. Since Eq. (8) does not require $R(\mathcal{N}_g)$ to be differentiable w.r.t. $g$, it can thus handle non-differentiable objectives. An interesting observation is that Eq. (8) has a similar form to the standard NAS (Zoph & Le, 2017), while it is not a sequential decision-making process and no RNN meta-controller is used in our case. Furthermore, since both gradient-based updates and REINFORCE-based updates are essentially two different update rules to the same binarized architecture parameters, it is possible to combine them to form a new update rule for the architecture parameters.

## 4 EXPERIMENTS AND RESULTS

We demonstrate the effectiveness of our proposed method on two benchmark datasets (CIFAR-10 and ImageNet) for the image classification task. Unlike previous NAS works (Zoph et al., 2018; Liu et al., 2018c) that first learn CNN blocks on CIFAR-10 under small-scale setting (e.g. fewer blocks), then transfer the learned block to ImageNet or CIFAR-10 under large-scale setting by repeatedly stacking it, we directly learn the architectures on the target task (either CIFAR-10 or ImageNet) and target hardware (GPU, CPU and mobile phone) while allowing each block to be specified.

### 4.1 EXPERIMENTS ON CIFAR-10

**Architecture Space.** For CIFAR-10 experiments, we use the tree-structured architecture space that is introduced by Cai et al. (2018b) with PyramidNet (Han et al., 2017) as the backbone[4]. Specifically,

---

[4]The list of operations in the candidate set is provided in the appendix.

| Model | Params | Test error (%) |
|---|---|---|
| DenseNet-BC (Huang et al., 2017) | 25.6M | 3.46 |
| PyramidNet (Han et al., 2017) | 26.0M | 3.31 |
| Shake-Shake + c/o (DeVries & Taylor, 2017) | 26.2M | 2.56 |
| PyramidNet + SD (Yamada et al., 2018) | 26.0M | 2.31 |
| ENAS + c/o (Pham et al., 2018) | 4.6M | 2.89 |
| DARTS + c/o (Liu et al., 2018c) | 3.4M | 2.83 |
| NASNet-A + c/o (Zoph et al., 2018) | 27.6M | 2.40 |
| PathLevel EAS + c/o (Cai et al., 2018b) | 14.3M | 2.30 |
| AmoebaNet-B + c/o (Real et al., 2018) | 34.9M | 2.13 |
| Proxyless-R + c/o (ours) | 5.8M | 2.30 |
| Proxyless-G + c/o (ours) | 5.7M | **2.08** |

Table 1: ProxylessNAS achieves state-of-the-art performance on CIFAR-10.

we replace all $3 \times 3$ convolution layers in the residual blocks of a PyramidNet with tree-structured cells, each of which has a depth of 3 and the number of branches is set to be 2 at each node (except the leaf nodes). For further details about the tree-structured architecture space, we refer to the original paper (Cai et al., 2018b). Additionally, we use two hyperparameters to control the depth and width of a network in this architecture space, i.e. $B$ and $F$, which respectively represents the number of blocks at each stage (totally 3 stages) and the number of output channels of the final block.

**Training Details.** We randomly sample 5,000 images from the training set as a validation set for learning architecture parameters which are updated using the Adam optimizer with an initial learning rate of 0.006 for the gradient-based algorithm (Section 3.2.1) and 0.01 for the REINFORCE-based algorithm (Section 3.3.2). In the following discussions, we refer to these two algorithms as **Proxyless-G** (gradient) and **Proxyless-R** (REINFORCE) respectively.

After the training process of the over-parameterized network completes, a compact network is derived according to the architecture parameters, as discussed in Section 3.2.1. Next, we train the compact network using the same training settings except that the number of training epochs increases from 200 to 300. Additionally, when the DropPath regularization (Zoph et al., 2018; Huang et al., 2016) is adopted, we further increase the number of training epochs to 600 (Zoph et al., 2018).

**Results.** We apply the proposed method to learn architectures in the tree-structured architecture space with $B = 18$ and $F = 400$. Since we do not repeat cells and each cell has 12 learnable edges, totally $12 \times 18 \times 3 = 648$ decisions are required to fully determine the architecture.

The test error rate results of our proposed method and other state-of-the-art architectures on CIFAR-10 are summarized in Table 1, where "c/o" indicates the use of Cutout (DeVries & Taylor, 2017). Compared to these state-of-the-art architectures, our proposed method can achieve not only lower test error rate but also better parameter efficiency. Specifically, Proxyless-G reaches a test error rate of 2.08% which is slightly better than AmoebaNet-B (Real et al., 2018) (the previous best architecture on CIFAR-10). Notably, AmoebaNet-B uses 34.9M parameters while our model only uses 5.7M parameters which is $6\times$ fewer than AmoebaNet-B. Furthermore, compared with PathLevel EAS (Cai et al., 2018b) that also explores the tree-structured architecture space, both Proxyless-G and Proxyless-R achieves similar or lower test error rate results with half fewer parameters. The strong empirical results of our ProxylessNAS demonstrate the benefits of directly exploring a large architecture space instead of repeatedly stacking the same block.

## 4.2 EXPERIMENTS ON IMAGENET

For ImageNet experiments, we focus on learning efficient CNN architectures (Iandola et al., 2016; Howard et al., 2017; Sandler et al., 2018; Zhu et al., 2018) that have not only high accuracy but also low latency on specific hardware platforms. Therefore, it is a multi-objective NAS task (Hsu et al., 2018; Dong et al., 2018; Elsken et al., 2018a; He et al., 2018; Wang et al., 2018; Tan et al., 2018), where one of the objectives is non-differentiable (i.e. latency). We use three different hardware platforms, including mobile phone, GPU and CPU, in our experiments. The GPU latency is measured on V100 GPU with a batch size of 8 (single batch makes GPU severely under-utilized). The CPU latency is measured under batch size 1 on a server with two 2.40GHz Intel(R) Xeon(R)

| Model | Top-1 | Top-5 | Mobile Latency | Hardware -aware | No Proxy | No Repeat | Search cost (GPU hours) |
|---|---|---|---|---|---|---|---|
| MobileNetV1 [16] | 70.6 | 89.5 | 113ms | - | - | ✗ | Manual |
| MobileNetV2 [30] | 72.0 | 91.0 | 75ms | - | - | ✗ | Manual |
| NASNet-A [38] | 74.0 | 91.3 | 183ms | ✗ | ✗ | ✗ | 48,000 |
| AmoebaNet-A [29] | 74.5 | 92.0 | 190ms | ✗ | ✗ | ✗ | 75,600 |
| MnasNet [31] | 74.0 | 91.8 | 76ms | ✓ | ✗ | ✗ | 40,000 |
| MnasNet (our impl.) | 74.0 | 91.8 | 79ms | ✓ | ✗ | ✗ | 40,000 |
| Proxyless-G (mobile) | 71.8 | 90.3 | 83ms | ✗ | ✓ | ✓ | 200 |
| Proxyless-G + LL | 74.2 | 91.7 | 79ms | ✓ | ✓ | ✓ | 200 |
| Proxyless-R (mobile) | **74.6** | **92.2** | 78ms | ✓ | ✓ | ✓ | 200 |

Table 2: ProxylessNAS achieves state-of-the art accuracy (%) on ImageNet (under mobile latency constraint $\leq 80ms$) with $200\times$ less search cost in GPU hours. "LL" indicates latency regularization loss. Details of MnasNet's search cost are provided in appendix C.

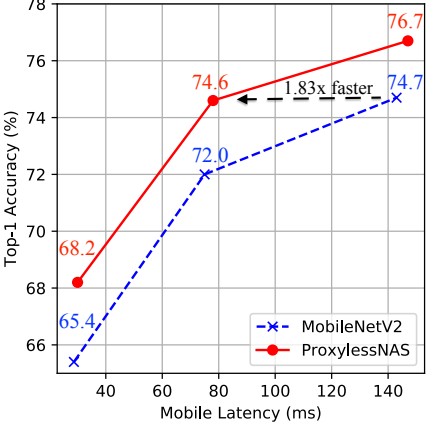

Figure 4: ProxylessNAS consistently outperforms MobileNetV2 under various latency settings.

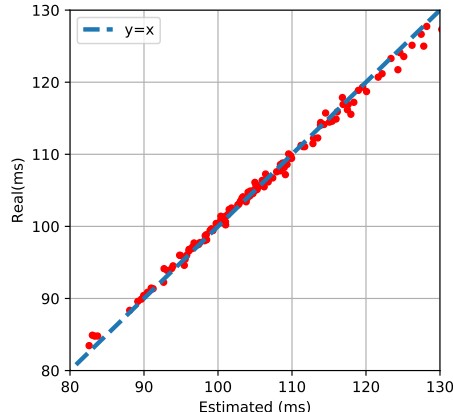

Figure 5: Our mobile latency model is close to $y = x$. The latency RMSE is 0.75ms.

CPU E5-2640 v4. The mobile latency is measured on Google Pixel 1 phone with a batch size of 1. For Proxyless-R, we use $ACC(m) \times [LAT(m)/T]^w$ as the optimization goal, where $ACC(m)$ denotes the accuracy of model $m$, $LAT(m)$ denotes the latency of $m$, $T$ is the target latency and $w$ is a hyperparameter for controlling the trade-off between accuracy and latency.

Additionally, on mobile phone, we use the latency prediction model (Appendix B) during architecture search. As illustrated in Figure 5, we observe a strong correlation between the predicted latency and real measured latency on the test set, suggesting that the latency prediction model can be used to replace the expensive mobile farm infrastructure (Tan et al., 2018) with little error introduced.

**Architecture Space.** We use MobileNetV2 (Sandler et al., 2018) as the backbone to build the architecture space. Specifically, rather than repeating the same mobile inverted bottleneck convolution (MBConv), we allow a set of MBConv layers with various kernel sizes $\{3, 5, 7\}$ and expansion ratios $\{3, 6\}$. To enable a direct trade-off between width and depth, we initiate a deeper over-parameterized network and allow a block with the residual connection to be skipped by adding the zero operation to the candidate set of its mixed operation. In this way, with a limited latency budget, the network can either choose to be shallower and wider by skipping more blocks and using larger MBConv layers or choose to be deeper and thinner by keeping more blocks and using smaller MBConv layers.

**Training Details.** We randomly sample 50,000 images from the training set as a validation set during the architecture search. The settings for updating architecture parameters are the same as CIFAR-10 experiments except the initial learning rate is 0.001. The over-parameterized network is trained on the remaining training images with batch size 256.

| Model | Top-1 | Top-5 | GPU latency |
|---|---|---|---|
| MobileNetV2 (Sandler et al., 2018) | 72.0 | 91.0 | 6.1ms |
| ShuffleNetV2 (1.5) (Ma et al., 2018) | 72.6 | - | 7.3ms |
| ResNet-34 (He et al., 2016) | 73.3 | 91.4 | 8.0ms |
| NASNet-A (Zoph et al., 2018) | 74.0 | 91.3 | 38.3ms |
| DARTS (Liu et al., 2018c) | 73.1 | 91.0 | - |
| MnasNet (Tan et al., 2018) | 74.0 | 91.8 | 6.1ms |
| **Proxyless (GPU)** | **75.1** | **92.5** | **5.1ms** |

Table 3: ImageNet Accuracy (%) and GPU latency (Tesla V100) on ImageNet.

**ImageNet Classification Results.** We first apply our ProxylessNAS to learn specialized CNN models on the mobile phone. The summarized results are reported in Table 2. Compared to MobileNetV2, our model improves the top-1 accuracy by 2.6% while maintaining a similar latency on the mobile phone. Furthermore, by rescaling the width of the networks using a multiplier (Sandler et al., 2018; Tan et al., 2018), it is shown in Figure 4 that our model consistently outperforms MobileNetV2 by a significant margin under all latency settings. Specifically, to achieve the same level of top-1 accuracy performance (i.e. around 74.6%), **MobileNetV2 has 143ms latency while our model only needs 78ms (1.83× faster)**. While compared with MnasNet (Tan et al., 2018), our model can achieve 0.6% higher top-1 accuracy with slightly lower mobile latency. More importantly, we are much more resource efficient: the GPU-hour is 200× fewer than MnasNet (Table 2).

Additionally, we also observe that Proxyless-G has no incentive to choose computation-cheap operations if were not for the latency regularization loss. Its resulting architecture initially has 158ms latency on Pixel 1. After rescaling the network using the multiplier, its latency reduces to 83ms. However, this model can only achieve 71.8% top-1 accuracy on ImageNet, which is 2.4% lower than the result given by Proxyless-G with latency regularization loss. Therefore, we conclude that it is essential to take latency as a direct objective when learning efficient neural networks.

Besides the mobile phone, we also apply our ProxylessNAS to learn specialized CNN models on GPU and CPU. Table 3 reports the results on GPU, where we find that our ProxylessNAS can still achieve superior performances compared to both human-designed and automatically searched architectures. Specifically, compared to MobileNetV2 and MnasNet, our model improves the top-1 accuracy by 3.1% and 1.1% respectively while being 1.2× faster. Table 4 shows the summarized results of our searched models on three different platforms. An interesting observation is that models optimized for GPU do not run fast on CPU and mobile phone, vice versa. Therefore, it is essential to learn specialized neural networks for different hardware architectures to achieve the best efficiency on different hardware.

**Specialized Models for Different Hardware.** Figure 6 demonstrates the detailed architectures of our searched CNN models on three hardware platforms: GPU/CPU/Mobile. We notice that the architecture shows different preferences when targeting different platforms: (i) The GPU model is shallower and wider, especially in early stages where the feature map has higher resolution; (ii) The GPU model prefers large MBConv operations (e.g. $7 \times 7$ MBConv6), while the CPU model would go for smaller MBConv operations. This is because GPU has much higher parallelism than CPU so it can take advantage of large MBConv operations. Another interesting observation is that our searched models on all platforms prefer larger MBConv operations in the first block within each stage where the feature map is downsampled. We suppose it might because larger MBConv operations are beneficial for the network to preserve more information when downsampling. Notably, such kind of patterns cannot be captured in previous NAS methods as they force the blocks to share the same structure (Zoph et al., 2018; Liu et al., 2018a).

## 5 CONCLUSION

We introduced ProxylessNAS that can directly learn neural network architectures on the target task and target hardware without any proxy. We also reduced the search cost (GPU-hours and GPU memory) of NAS to the same level of normal training using path binarization. Benefiting from the direct search, we achieve strong empirical results on CIFAR-10 and ImageNet. Furthermore,

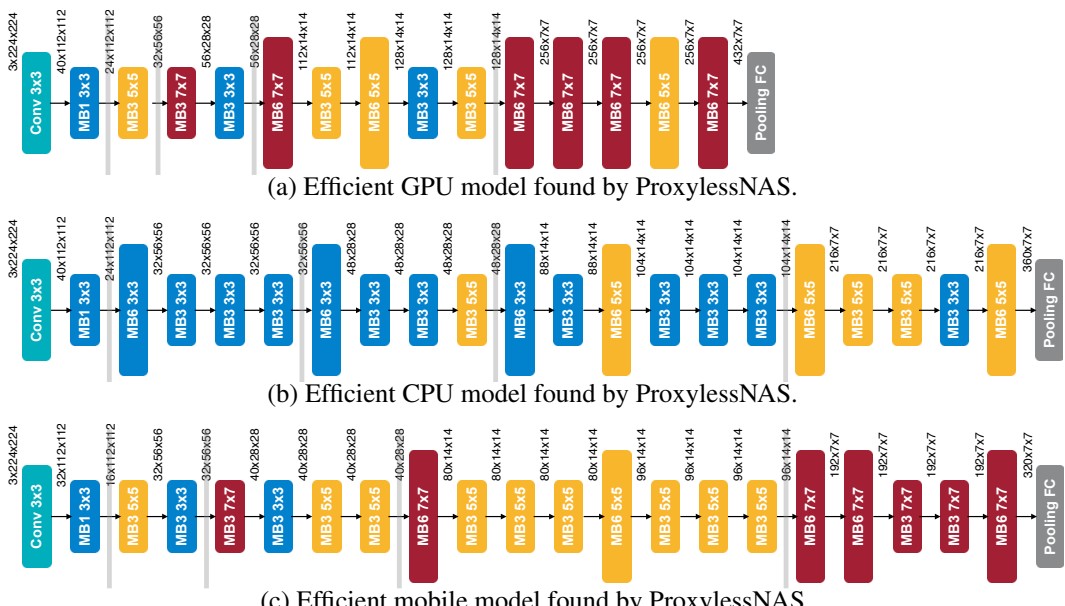

(a) Efficient GPU model found by ProxylessNAS.

(b) Efficient CPU model found by ProxylessNAS.

(c) Efficient mobile model found by ProxylessNAS.

Figure 6: Efficient models optimized for different hardware. "MBConv3" and "MBConv6" denote mobile inverted bottleneck convolution layer with an expansion ratio of 3 and 6 respectively. Insights: GPU prefers shallow and wide model with early pooling; CPU prefers deep and narrow model with late pooling. Pooling layers prefer large and wide kernel. Early layers prefer small kernel. Late layers prefer large kernel.

| Model | Top-1 (%) | GPU latency | CPU latency | Mobile latency |
|---|---|---|---|---|
| Proxyless (GPU) | 75.1 | 5.1ms | 204.9ms | 124ms |
| Proxyless (CPU) | 75.3 | 7.4ms | 138.7ms | 116ms |
| Proxyless (mobile) | 74.6 | 7.2ms | 164.1ms | 78ms |

Table 4: Hardware prefers specialized models. Models optimized for GPU does not run fast on CPU and mobile phone, vice versa. ProxylessNAS provides an efficient solution to search a specialized neural network architecture for a target hardware architecture, while cutting down the search cost by $200\times$ compared with state-of-the-arts (Zoph & Le, 2017; Tan et al., 2018).

we allow specializing network architectures for different platforms by directly incorporating the measured hardware latency into optimization objectives. We compared the optimized models on CPU/GPU/mobile and raised the awareness of the needs of specializing neural network architecture for different hardware architectures.

ACKNOWLEDGMENTS

We thank MIT Quest for Intelligence, MIT-IBM Watson AI lab, SenseTime, Xilinx, Snap Research for supporting this work. We also thank AWS Cloud Credits for Research Program providing us the cloud computing resources.

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

## A    THE LIST OF CANDIDATE OPERATIONS USED ON CIFAR-10

We adopt the following 7 operations in our CIFAR-10 experiments:

- $3 \times 3$ dilated depthwise-separable convolution
- Identity
- $3 \times 3$ depthwise-separable convolution
- $5 \times 5$ depthwise-separable convolution
- $7 \times 7$ depthwise-separable convolution
- $3 \times 3$ average pooling
- $3 \times 3$ max pooling

## B    MOBILE LATENCY PREDICTION

Measuring the latency on-device is accurate but not ideal for scalable neural architecture search. There are two reasons: (i) *Slow.* As suggested in TensorFlow-Lite, we need to average hundreds of runs to produce a precise measurement, approximately 20 seconds. This is far more slower than a single forward / backward execution. (ii) *Expensive.* A lot of mobile devices and software engineering work are required to build an automatic pipeline to gather the latency from a mobile farm. Instead of direct measurement, we build a model to estimate the latency. We need only 1 phone rather than a farm of phones, which has only 0.75ms latency RMSE. We use the *latency model* to search, and we use the *measured latency* to report the final model's latency.

We sampled 5k architectures from our candidate space, where 4k architectures are used to build the latency model and the rest are used for test. We measured the latency on Google Pixel 1 phone using TensorFlow-Lite. The features include (i) type of the operator (ii) input and output feature map size (iii) other attributes like kernel size, stride for convolution and expansion ratio.

## C    DETAILS OF MNASNET'S SEARCH COST

Mnas (Tan et al., 2018) trains 8,000 mobile-sized models on ImageNet, each of which is trained for 5 epochs for learning architectures. If these models are trained on V100 GPUs, as done in our experiments, the search cost is roughly 40,000 GPU hours.

## D    IMPLEMENTAION OF THE GRADIENT-BASED ALGORITHM

A naive implementation of the gradient-based algorithm (see Eq. (4)) is calculating and storing $o_j(x)$ in the forward step to later compute $\partial L/\partial g_j$ in the backward step:

$$\partial L/\partial g_j = \text{reduce\_sum}(\nabla_y L \circ o_j(x)), \tag{9}$$

where $\nabla_y L$ denotes the gradient w.r.t. the output of the mixed operation $y$, "$\circ$" denotes the element-wise product, and "reduce_sum($\cdot$)" denotes the sum of all elements.

Notice that $o_j(x)$ is only used for calculating $\partial L/\partial g_j$ when $j^{th}$ path is not active (i.e. not involved in calculating $y$). So we do not need to actually allocate GPU memory to store $o_j(x)$. Instead, we can calculate $o_j(x)$ after getting $\nabla_y L$ in the backward step, use $o_j(x)$ to compute $\partial L/\partial g_j$ following Eq. (9), then release the occupied GPU memory. In this way, without the approximation discussed in Section 3.2.1, we can reduce the GPU memory cost to the same level of training a compact model.

