# OpenReview forum: "ProxylessNAS: Direct Neural Architecture Search on Target Task and Hardware"
_ICLR.cc/2019/Conference_

### Official Review · AnonReviewer1 · 2018-11-02
**Interesting idea for efficient NAS that gives state-of-the-art results (on limited datasets)**

**Rating:** 7
**Confidence:** 2

**Review:**

The algorithm described in this paper is part of the one-shot family of architecture search algorithms. In practice this means training an over-parameterized architecture, of which the architectures being searched for are sub-graphs. Once this bigger network is trained it is pruned into the desired sub-graph. The algorithm is similar to DARTS in that it it has weights that determine how important the various possible nodes are, but the interpretation here is stochastic, in that the weight indicates the probability of the component being active. Two methods to train those weights are being suggested, using REINFORCE and using BinaryConnect, both having different trade offs.

- (minor) *cumbersome* network seems the wrong term, maybe over-parameterized network?
- (minor) I do not think that the size of the search space a very meaningful metric

Pros:
- Good exposition
- Interesting and fairly elegant idea
- Good experimental results

Cons
- tested on a limited amount of settings, for something that claims that helps to automate the creation of architecture. I think this is the main shortcoming, although shared by many NAS papers
- No source code available

Some typos:

- Fo example, when proxy strategy -> Fo*r* example
- normal training in following ways. -> in *the* following ways
- we can then derive optimized compact architecture.

---

> ### Author Response · Authors · 2018-11-10
> **Proxyless NAS enables efficient and direct search on different tasks and hardware platforms**
>
> We sincerely thank you for the detailed comments on our paper. We have revised the paper and fixed the typos accordingly.
>
> >>> Response to “limited amount of tested settings”:
> As our proxy-less NAS has reduced the cost to the same level of normal training (100x more efficient on ImageNet), it is of great interest for us to apply proxy-less NAS to more settings and datasets. However, for this work, considering the resource constraints and time limits, we have strong reasons to believe that our experiment settings are sufficient:
>
> a) Our experiments are conducted on two most representative benchmarks (CIFAR and ImageNet). It is in line with previous NAS papers and also makes it possible to compare our method with previous NAS methods. We also experimented with 3 different hardware platforms and observed consistent latency improvement over previous work.
>
> b) Moreover, on the challenging ImageNet classification task, we have conducted architecture search experiments under three different settings (GPU, CPU and Mobile) while previous NAS papers mainly transfer learned architectures from CIFAR-10 to ImageNet without conducting architecture search experiments on ImageNet [1, 2].
>
> >>> Response to “no source code available”:
> Reviewer 2 also has similar requests, based on the concern on our strong empirical results. Our pre-trained models and the evaluation code are provided in the following anonymous link: https://goo.gl/QU3GhA. Besides, we have also uploaded the video visualizing the architecture search process: https://goo.gl/VAzGJs. We plan to open source our project upon publication.
>
> >>> Response to “the size of the search space is not a very meaningful metric”:
> This might be a misunderstanding. We do not intend to use the size of our search space as a metric for comparison; instead, it is an important reason why our accuracy is much better than previous NAS methods. Previous NAS methods forced different blocks to share the same structure and only explored a limited architecture space (e.g. 10^18 in [2] and 10^10 in [3]). Our method, breaking the constraints, allows all of the blocks to be specified and has much larger search space (i.e. 10^547).
>
> [1] Zoph B, Vasudevan V, Shlens J, Le QV. Learning transferable architectures for scalable image recognition. CVPR 2018.
> [2] Liu H, Simonyan K, Yang Y. Darts: Differentiable architecture search. arXiv preprint arXiv:1806.09055. 2018.
> [3] Bender G, Kindermans PJ, Zoph B, Vasudevan V, Le Q. Understanding and simplifying one-shot architecture search. ICML 2018.

---

> > ### Comment · AnonReviewer1 · 2018-11-26
> > **Thank you for your response**
> >
> > Thank you for your response. I particularly appreciate the release of the source code, while I did not have time to dig into it, it definitely increases the trust from the reader.
> >
> > Regarding the limited experiments, consider it a criticism towards the sub-field in general, not to this paper in particular. It just seems a bit counter to the narrative of automatically selecting architectures if only a very limited amount of architectures are found.
> >
> > I do appreciate how this paper is searching a slightly more varied architecture search compared to some previous methods, but I do not think the search space absolute size (10^547) says much in this regard, it would be easy to artificially come up with large search spaces with little variety as well as small search spaces with a lot of variety. My personal opinion is that it would be better to omit the number, mist giving the impression that it has more meaning than it has, but consider it a very minor point :)

---

> > > ### Author Response · Authors · 2018-11-29
> > > **Thanks for your further feedback. We have revised the paper accordingly.**
> > >
> > > Thank you for your reply and detailed suggestion. We have uploaded a revision of our paper and removed the number of search space size.

---

### Official Review · AnonReviewer2 · 2018-11-03
**Solid work with convincing results**

**Rating:** 6
**Confidence:** 2

**Review:**

It seems the authors propose an efficient method to search platform-aware network architecture aiming at high recognition accuracy and low latency. Their results on CIFAR-10 and ImageNet are surprisingly good.  But it is still hard to believe that the author can  achieve 2.08% error rate with only 5.7M parameter on CIFAR10 and 74.5% top-1 accuracy on ImageNet with less GPU hours/memories than prior arts.

Given my concerns above, the author must release their code and detail pipelines since NAS papers are difficult to be reproduced.

There is a small typo in reference part:
Jing-Dong Dong's work should be DPP-Net instead of PPP-Net (https://eccv2018.org/openaccess/content_ECCV_2018/papers/Jin-Dong_Dong_DPP-Net_Device-aware_Progressive_ECCV_2018_paper.pdf)
and I think this paper "Neural Architecture Optimization" shoud be cited.

---

> ### Author Response · Authors · 2018-11-10
> **Models on all platforms have been open sourced. Reproducible experiment verified on 3 different platforms.**
>
> We sincerely thanks for the detailed feedback. Our pre-trained models and the evaluation code are provided in the following anonymous link for verifying our results: https://goo.gl/QU3GhA. We have also made a video to visualize the architecture search process: https://goo.gl/VAzGJs. We would like to release the entire codebase upon publication.
>
> >>> Response to “performances are too good to be true”:
> We consider the comment as a compliment rather than a drawback. There are several reasons for our good results:
> a) Our proxy-less NAS *directly* learns on the *target* task while previous NAS methods *indirectly* learn on *proxy* tasks. For example, on CIFAR-10, DARTS [1] conducted architecture search experiments with 8 blocks due to their high memory consumption and then transferred the learned block structure to a much larger network with 20 blocks. This indirect optimization scheme would lead to suboptimal results while our proxy-less NAS does not suffer from this problem.
>
> b) We broke the convention in neural architecture design by *not* repeating the same building block structure. Our method explores a much larger architecture space compared to previous NAS methods (10^547 vs 10^18). Furthermore, our method has much larger block diversity and is able to learn preferences at different positions in the architecture.
>
> For example, our optimized neural network architectures for GPU, CPU and mobile phone prefer to choose more computation-expensive operations (e.g. 7x7 MBConv6) for the last few stages where the resolution of feature map is low. They also prefer to choose more computation-expensive operations in the first block within each stage where the feature map is downsampled. We consider the ability to learn such patterns which are absent in previous NAS papers also helps to improve our results.
>
> >>> Response to “DPP-Net and NAO citations”:
> Apologize for the typo and missing a relevant paper in our reference part. We have fixed typo and added a reference to “Neural Architecture Optimization”. Thanks for pointing out our mistakes.
>
> [1] Liu H, Simonyan K, Yang Y. Darts: Differentiable architecture search. arXiv preprint arXiv:1806.09055. 2018.

---

### Official Review · AnonReviewer3 · 2018-11-04
**Interesting combination of existing methods and good performance**

**Rating:** 6
**Confidence:** 4

**Review:**


This paper addresses the problem of architecture search, and specifically seeks to do this without having to train on "proxy" tasks where the problem is simplified through more limited optimization, architectural complexity, or dataset size. The paper puts together a set of existing complementary methods towards this end, specifically 1) Training "cumbersome" networks as in One Shot and DARTS, 2) Path binarization to address memory requirements (optimized using ideas in BinaryConnect), and 3) optimizing a non-differentiable architecture using REINFORCE. The end result is that this method is able to find efficient architectures that achieve state of art performance with fewer parameters, can be optimized for non-differentiable objectives such as latency, and can do so with smaller amounts of GPU memory and computation.

Strengths

 + The paper is in general well-written and provides a clear description of the methods.

 + Different choices made are well-justified in terms of the challenge they seek to address (e.g. non-differentiable objectives, etc.)

 + The results achieve state of art while being able to trade off other objectives such as latency

 + There are some interesting findings such as the need for specialized blocks rather than repeating blocks, comparison of architectures for CPUs vs. GPUs, etc.

Weaknesses

 - In the end, the method is really a combination of existing methods (One Shot/DART, BinaryConnect, use of RL/REINFORCE, etc.). One novel aspect seems to be factorizing the choice out of N candidates by making it a binary selection. In general, it would be good for the paper to make clear which aspects were already done by other approaches (or if it's a modification what exactly was modified/added in comparison) and highlight the novel elements.

 - The comparison with One Shot and DARTS seems strange, as there are limitations place on those methods (e.g. cell structure settings) that the authors state they chose "to save time". While that consideration has some validity, the authors should explicitly state why they think these differences don't unfairly bias the experiments towards the proposed approach.

 - It's not clear that the REINFORCE aspect is adding much; it achieves slightly higher parameters when compared against Proxyless-G, and while I understand the motivation to optimize a non-differentiable function in this case the latency example (on ImageNet) is never compared to Proxyless-G. It could be that optimized the normal differentiable objective achieves similar latency with the smaller number of parameters. Please show results for Proxyless-G in Table 4.

 - There were several typos throughout the paper ("great impact BY automatically designing", "Fo example", "is build upon", etc.)

 In summary, the paper presents work on an interesting topic. The set of methods seem to be largely pulled from work that already exists, but is able to achieve good results in a manner that uses less GPU memory and compute, while supporting non-differentiable objectives. Some of the methodological issues mentioned above should be addressed though in order to strengthen the argument that all parts of the the method (especially REINFORCE) are necessary.

---

> ### Author Response · Authors · 2018-11-10
> **proxy-less NAS is an important contribution that breaks many conventions and stereotypes of neural architecture design.  It's not a combination of existing methods.**
>
> We sincerely thank you for your comprehensive comments and constructive advices.
>
> >>> Response to “combination of existing methods”:
> Thanks for your kind advice on organizing the paper to make our contributions more clear. Here, we would like to emphasize our contributions:
>
> a) Our proxy-less NAS is the first NAS algorithm that directly learns architectures on the large-scale dataset (e.g. ImageNet) without any proxy. We also solved an important problem improving the computation efficiency of NAS as we reduced the computational cost (GPU hours and GPU memory) of NAS to the same level as normal training. Moreover, the GPU memory requirement of our method keeps at O(1) complexity rather than grows linearly with the number of candidate operations O(N)  [3, 4]. Therefore, our method can easily support a large candidate set while DARTS and One-Shot cannot.
>
> b) Our proxy-less NAS is the first NAS algorithm that breaks the convention of repeating blocks in neural architecture design. From Alexnet and VGG to ResNet and MobileNet, manually designed CNNs used to repeat blocks within the same stage. Previous NAS works keep the tradition as otherwise the searching cost will be unaffordable. Our work breaks the constraints, and we found this is actually a stereotype that needs to be corrected.
>
> The new interesting design patterns, found by our method, can provide new insights for efficient neural architecture design. For example, people used to stack multiple 3x3 convs to replace a single large kernel conv, as this uses fewer parameters while keeping a similar receptive field. But we found this pattern may not be proper for designing efficient (low latency) networks: Two 3x3 depthwise separable convs actually run slower than a single 5x5 depthwise separable conv.  Our GPU model, shown in Figure 4, incorporates large kernel convs and aggressively pools at early stages to shrink network depth. Then the model chooses computation-expensive operations at low-resolution stages. It also tends to choose computation-expensive operations in the first block within each stage where the feature map is downsampled.  As a consequence, our GPU model can outperform previous SOTA efficient architectures in accuracy performances (e.g. 3.1% higher top-1 than MobileNetV2), while running faster than them (e.g. 1.2x faster than MobileNetV2). Such patterns cannot be found by previous NAS, as they optimize on proxy task and force blocks to share structures.
>
> c) Our method builds upon methods from two communities (one-shot architecture search from NAS community and Pruning/BinaryConnect from model compression community). It is the first time to incorporate ideas from the model compression community to the NAS community and we also provide a new path-level pruning perspective for one-shot architecture search. Moreover, we provide a unified framework for both gradient-based updates and REINFORCE-based updates.
>
> d) Our proxy-less NAS achieved very strong empirical results on two most representative benchmarks (i.e. CIFAR and ImageNet). On CIFAR-10, our optimized model reached 2.08% error rate with only 5.7M parameters, outperforming previous state-of-the-art architecture (AmeobaNet-B with 34.9M parameters). On ImageNet, we searched specialized neural network architectures for three different platforms (GPU, CPU and mobile phone). With latency constraints, our optimized models also achieved state-of-the-art results (3.1% higher top-1 accuracy while being 1.2x faster on GPU and 2.6% higher top-1 accuracy with similar latency on mobile phone, compared to MobileNetV2).
>
> Besides, we directly optimize the latency, rather than an inaccurate proxy (i.e. FLOPs). It’s an important concept that low FLOPs doesn’t translate to low latency. All our speedup numbers are reported with real measured latency. We believe both our efficient search methodology and the resulting efficient models have big industry impact.

---

> > ### Author Response · Authors · 2018-11-10
> > **We made Apple-to-Apple comparison. Our advantage on memory saving is clear.**
> >
> >
> > >>> Response to “comparison with One Shot and DARTS”:
> > Apologize for the unclear explanation for this experiment. We will revise this part to make it more clear.
> >
> > All of three methods are evaluated under the same condition except DARTS [3]. Same as the original paper, DARTS *has to* use a smaller scale setting for learning architectures due to the high memory consumption. So for DARTS, the first cell structure setting is chosen to fit the network into a single GPU to learn cell structure. Then we evaluated the learned cell structure on two larger settings by repeatedly stacking it, same as the original DARTS paper [3].
> >
> > For our method, since we solved the high memory consumption issue via binarized path, our method can directly learn architectures under both small-scale and large-scale settings with *limited* GPU memory. As it is one of the key advantages of our method over previous NAS methods, we consider it reasonable to keep such differences.
> >
> > >>> Response to “add results for Proxyless-G on ImageNet”:
> > Thanks for suggesting this new experiment. We have launched this experiment and will add the results to the paper.
> >
> > However, it is important to take latency as a *direct* objective when learning specialized neural network architectures for a platform. Otherwise, NAS would fail to make a good trade-off between accuracy and latency. For example, NASNet-A [1] and AmoebaNet-A [2] has shown compelling accuracy results compared to MobileNetV2 1.4 with similar number of parameters and FLOPs. But they are optimized without the awareness of the latency, their measured latencies on mobile phone are much worse than MobileNetV2 1.4 (see below). Therefore, we employ REINFORCE to directly optimize the non-differentiable objective (i.e. latency).
> >
> > Model				Params	        FLOPS	        Top-1	Mobile latency
> > MobileNet V2 1.4		6.9M		585M		74.7		143ms
> > NASNet-A			5.3M		564M		74.0		183ms
> > AmeobaNet-A		5.1M		555M		74.5		190ms
> >
> > [1] Zoph B, Vasudevan V, Shlens J, Le QV. Learning transferable architectures for scalable image recognition. CVPR 2018.
> > [2] Real E, Aggarwal A, Huang Y, Le QV. Regularized evolution for image classifier architecture search. arXiv preprint arXiv:1802.01548. 2018.
> > [3] Liu H, Simonyan K, Yang Y. Darts: Differentiable architecture search. arXiv preprint arXiv:1806.09055. 2018.
> > [4] Bender G, Kindermans PJ, Zoph B, Vasudevan V, Le Q. Understanding and simplifying one-shot architecture search. ICML 2018.

---

> > > ### Author Response · Authors · 2018-11-20
> > > **We have added the results for Proxyless-G on ImageNet. And we also include a new differentiable approach to handle non-differentiable objectives (i.e. latency).**
> > >
> > > We have added the results for Proxyless-G on ImageNet to the paper (please see Table 6 in Appendix D). We find that without taking latency as a direct objective, Proxyless-G has no incentive to choose computation-cheap operations. Consequently, it designs a very slow network that has 158ms latency on mobile phone. After rescaling the network using depth multiplier [1, 2], the latency of the network reduces to 83ms. However, this model can only achieve 71.8% top-1 accuracy on ImageNet which is 2.8% lower than Proxyless-R. Therefore, as discussed in our previous responses, it is essential to take latency which is non-differentiable as a direct optimization objective. And REINFORCE-based approach provides a solution to this problem.
> > >
> > > Beside REINFORCE, we have recently designed a differentiable approach to handle the non-differentiable objectives (please see Appendix D). Specifically,  we propose the latency regularization loss based on our proposed latency prediction model (please see Appendix C). The key to the latency regularization loss is an observation that the expected latency of a mixed operation is actually differentiable w.r.t. architecture parameters. Therefore, by incorporating the expected latency into the loss function as a regularization term, we are able to directly optimize the trade-off between accuracy and latency. Further details are provided in Appendix D.
> > >
> > > [1] Sandler, Mark, et al. "MobileNetV2: Inverted Residuals and Linear Bottlenecks." Proceedings of the IEEE Conference on Computer Vision and Pattern Recognition. 2018.
> > > [2] Tan, Mingxing, et al. "Mnasnet: Platform-aware neural architecture search for mobile." arXiv preprint arXiv:1807.11626 (2018).

---

> > ### Comment · AnonReviewer3 · 2018-11-26
> > **Thank you for the response**
> >
> >
> >  Thanks for the detailed response. Please see comments below.
> >
> > > a) Our proxy-less NAS is the first NAS algorithm that directly learns architectures on the large-scale dataset (e.g. ImageNet) without any proxy.
> >
> > I agree but this is not a method/algorithmic contribution but an empirical one. The way you achieve this is by combining existing methods (which I listed in the original review), which allows the reduction of memory usage/computation compared to One-Shot/DART. I should emphasize that there is nothing particularly wrong with combining methods (especially across areas/fields) but just makes the empirical contribution and thoroughness of the analysis more important. However, the method/algorithmic contributions should be made clear in a precise manner, rather than making large general statements.
> >
> > > b) Our proxy-less NAS is the first NAS algorithm that breaks the convention of repeating blocks in neural architecture design.
> >
> >   I am not sure this is the case. Neuroevolution methods (which you should cite more heavily) do not necessarily require this, e.g. [1]. However, I agree that within the regime of training over-parameterized networks or methods scalable. Again, please state your advantages explicitly; you seem to mention one axis/dimension at a time (e.g. scalability, no proxy, no repeating cell structure) yet your advantages are really at the combination of these. It would be nice to actually have a table showing the strengths/weaknesses along these axes for all of these methods, which would make it more clear.
> >
> > [1] Large-Scale Evolution of Image Classifiers, Esteban Real, Sherry Moore, Andrew Selle, Saurabh Saxena, Yutaka Leon Suematsu, Jie Tan, Quoc Le, Alex Kurakin, https://arxiv.org/abs/1703.01041
> >
> > > The new interesting design patterns, found by our method, can provide new insights for efficient neural architecture design.
> >
> > I agree with this and mentioned it in the review.
> >
> > > c) Our method builds upon methods from two communities (one-shot architecture search from NAS community and Pruning/BinaryConnect from model compression community).
> >
> > Again, I agree but this means that it *is* a combination of methods (which contradicts your rebuttal title).
> >
> > > With latency constraints, our optimized models also achieved state-of-the-art results (3.1% higher top-1 accuracy while being 1.2x faster on GPU and 2.6% higher top-1 accuracy with similar latency on mobile phone, compared to MobileNetV2).
> > > Besides, we directly optimize the latency, rather than an inaccurate proxy (i.e. FLOPs).
> >
> > I agree it's interesting to optimize for these non-differentiable objectives. However, it seems to me that given that you are optimizing directly for them, the actual gains are not that large. For example, in the new mobile phone results you have presented there is a network that actually has better latency with slightly worse accuracy, which makes it hard to compare:
> >
> > MobileNet V2		72.0		91.0		75ms
> > Proxyless NAS (ours)	74.6		92.2		78ms
> >
> > In all, it would be great for the authors to precisely define what is novel about the method (if it is not a combination of existing methods, as you claim in the rebuttal title). If it is a combination of methods (which again should not necessarily be seen as a bad thing), then it would be great to emphasize exactly the empirical contribution (the largest of which seems to be the reduction of memory/compute for training of large over-parameterized networks, scaled to ImageNet-sized datasets). The optimization of a non-differentiable objective can also be a smaller contribution, but is common to RL-based methods. Again, I think this paper presents some nice results, but it is important to be precise and not make more general claims than warranted.

---

> > > ### Author Response · Authors · 2018-11-29
> > > **Thanks for your helpful feedback.**
> > >
> > > Thank you for your helpful feedback. We have revised our paper according to your suggestion.
> > >
> > > >>> “in the new mobile phone results you have presented there is a network that actually has better latency with slightly worse accuracy, which makes it hard to compare”
> > >
> > > 2.6% top-1 accuracy improvement on ImageNet is significant. To achieve the same accuracy, MobileNetV2 needs 2x latency (143ms v.s. 78ms). Please see Figure 4.
> > >
> > > >>> “It would be nice to actually have a table showing the strengths/weaknesses along these axes for all of these methods”
> > >
> > > Thanks for your suggestion. We will add the table to our paper.
> > >
> > > Model	                             Top-1	  Top-5	Latency	Hardware-Aware	  No-Proxy	No-Repeating	Time	Memory
> > > MobilenetV1	                      70.6	   89.5	 113ms	              -	                         -	                  No	                    -	               -
> > > MobilenetV2	                      72.0	   91.0	  75ms	              -	                         -	                  No	                    -                  -
> > > NASNet-A	                      74.0	   91.3	 183ms	            No	               No                      No                  10^4  	   10^1
> > > AmoebaNet-A	              74.5	   92.0	 190ms	            No	               No	                  No	                 10^4         10^1
> > > Darts	                              73.1	   91.0	      -	                    No	               No	                  No	                 10^2	   10^2
> > > MnasNet	                      74.0	   91.8	  79ms	            Yes	               No	                  No	                 10^4    	  10^1
> > > ProxylessNAS (mobile)      74.6	   92.2	  78ms	            Yes	               Yes	                  Yes                 10^2    	  10^1
> > >
> > > >>> “precisely define what is novel about the method” and “emphasize exactly the empirical contribution”
> > >
> > > We summarize our contributions as follows:
> > >
> > > > Methodologically,
> > > a) We provided a new path-level pruning perspective for NAS.
> > >
> > > b) We proposed a gradient-based approach (Section 3.3.1) to handle non-differentiable hardware objectives (e.g. latency), making them differentiable by introducing regularization loss.
> > >
> > > c) We proposed a path-level binarization approach to address the high memory consumption issue of differentiable NAS. Notably, different from BinaryConnect that binarizes each weight, our path-level binarization approach binarizes the entire path.
> > >
> > > > Empirically,
> > > a) We significantly reduced the cost of memory/compute for the training of large over-parameterized networks and thereby scaled to large-scale datasets (ImageNet) without proxy and repeating blocks.
> > >
> > > b) We studied specialized neural network architectures for different hardware architectures and showed its advantage, raising people’s awareness of specializing neural network architectures for hardware.
> > >
> > > c) We achieved strong empirical results on both CIFAR-10 and ImageNet. On different hardware platforms (GPU, CPU and mobile phone), our models not only significantly outperform previous state-of-the-arts, but also peer submissions.
> > >
> > > We sincerely thank your feedback and hopefully have cleared your concerns.

---

### Author Response · Authors · 2018-11-02
**New experiment results on mobile phone**

Hi all,

Our efficient algorithm allows us to specialize neural network architectures for different devices easily. Recently, we extended our proxyless NAS to the mobile setting and achieved SOTA result with mobile latency constraint (< 80ms latency on Pixel 1 phone) as well. The following is our current results on ImageNet (Device: Pixel 1. Batch size: 1. Framework: TF-Lite):

Model				Top-1	Top-5	Mobile latency
MobileNet V1		70.6		89.5		113ms
MobileNet V2		72.0		91.0		75ms
NASNet-A			74.0		91.3		183ms
AmeobaNet-A		74.5		92.0		190ms
MnasNet			74.0		91.8		76ms
MnasNet (our impl.)	74.0		91.8		79ms
Proxyless NAS (ours)	74.6		92.2		78ms

The detailed architectures of our searched models and their learning process are provided in the following anonymous link:
https://drive.google.com/open?id=1nut1owvACc9yz1ZPqcbqoJLS2XrVPp1Q

---

### Public Comment · (anonymous) · 2018-11-09
**can you release code?**

Dear authors, can you release your source code for readers to validate your experiment?

---

> ### Author Response · Authors · 2018-11-09
> **We have uploaded the evaluation code and pretrained models**
>
> Thanks for your interest in our work. The evaluation code and pretrained models are accessible at https://goo.gl/QU3GhA. We also made a video to visualize the architecture search process at https://goo.gl/VAzGJs . You are welcome to validate the performance. The entire codebase will be released upon publication.
>
> Our implementation is repeatable and reproducible. We used the same code base to search CPU/GPU/Mobile models. On all three platforms the performance consistently outperformed previous work, thanks to our Proxyless NAS enables searching over a large design space efficiently.

---

> > ### Public Comment · (anonymous) · 2018-12-19
> > **Only releasing training code is meaningful**
> >
> > Dear the authors,
> >
> > I want to echo with the reviewers/public readers that releasing your detailed training pipeline is quite crucial given the good performances reported in the paper. Furthermore, only evaluation code/model ckpts is definitely not enough since people have various unreasonable ways to obtain a good ckpt only on the test set (I'm not meaning you are doing this and sorry for possible offense here in advance).
> >
> > Best

---

> > ### Public Comment · (anonymous) · 2019-02-08
> > **Code for training**
> >
> > Dear Authors,
> >
> > As the paper has now been accepted, I kindly request you to release the training code for the paper. Thank you.

---

### Public Comment · ~Robin_Tibor_Schirrmeister1 · 2018-11-13
**Implementation question regarding rescaling of architecture parameters**

Thanks for the fascinating research work.
I am trying to reimplement your method and have a question regarding:
"Finally,  as path weights are computed by applying softmax to the architecture parameters, we need to rescale the value of these two updated architecture parameters by multiplying a ratio to keep the path weights of unsampled paths unchanged."

I am not sure how to do this correctly, can you provide the formula for this ratio or code? I am a bit stuck there, how to compute the ratio :)

Another question regarding:
"Following this idea, within an update step of the architecture parameters, we first sample two paths according to the multinomial distribution (p1,···,pN) and mask all the other paths as if they do not exist."

Could this sampling result in the same path being chosen twice? And do you handle that in some way?

---

> ### Public Comment · ~Robin_Tibor_Schirrmeister1 · 2018-11-14
> **Further questions**
>
> Let me expand a little bit on the question and just write my understanding and open questions regarding the Gradient-Based Updates from section 3.1.
>
> So, given a_i's as architecture weights, I am implementing it as follows:
> 1. Compute p_i's from a_i's using softmax
> 2. Use computed p_i's as sampling probabilities for the multinomial distribution to select two operations. [Possibly resample, if same operation chosen twice?]
> 3. Recompute p_i's of the chosen a_i's by only pushing the two chosen a_is through softmax? Let's call them pnew_i's
> 4. Use pnew_i's as input to binarize function, which will select one operation as active and one as inactive
> 5. Compute outputs for both chosen operations, let's call them o_1, o_2, with o_1 the active operation according to the binarize function computed before
> 6. Compute overall output as g_1(=1)*o_1 + g_2(=0)*o_2 (g_1, g_2 from binarize)
> 7. Compute gradient on chosen a_i's as (gradient of loss wrt g_i) * (gradient of pnew_i wrt a_i) [or using full softmax, i.e. (gradient of loss wrt g_i) * (gradient of p_i wrt a_i)?]
> 8. Make update step on a_i's with optimizer
> 9. Multiply updated and chosen a_is by a factor that keeps probabilities p_is of unchosen operations identical to before [or see update below]
>
> What is correct, what is not?
>
> Also, you use Adam for the architecture parameters, do you think it can be a problem for the adaptive gradient averages that in a single update, most operations are not chosen? Or do you sample multiple times before you make an Adam update step?

---

> > ### Public Comment · ~Robin_Tibor_Schirrmeister1 · 2018-11-15
> > **Rescaling code**
> >
> > So, on further thought I assume you might have meant rescaling probabilities of sampled operations by a factor such that probabilities of  unsampled operations stay the same. And update the corresponding alphas for the sampled operations such that this matches.
> >
> > I have tried to do this here:
> > https://gist.github.com/robintibor/83064d708cdcb311e4b453a28b8dfdca
> >
> > Does this look correct to you?

---

> ### Author Response · Authors · 2018-11-19
> **Responses to implementation questions**
>
> Hi Robin,
>
> Thanks for your interest in our work and your detailed questions.
>
> >>> Response to "Rescaling architecture parameters"
> Your understanding of the gradient-based updates is correct.
> As for sampling two paths according to the multinomial distribution, we use "torch.multinomial()". And by setting "replacement=False", the same path will not be chosen twice.
>
> >>> Response to "Adam optimizer for architecture parameters"
> We also consider it would be problematic to use the adaptive gradient averages for this case where most of the paths are not chosen. So we set beta1 to be 0 in the Adam optimizer.  Sampling multiple times before making an Adam update step is a nice idea. We will try it later. Thanks for your suggestion.

---

> > ### Public Comment · ~Robin_Tibor_Schirrmeister1 · 2018-11-19
> > **replacement=True or False?**
> >
> > Thanks for your answers! I assume you man replacement=False, right?
> > beta1 is set zero, and what value do you use for beta2?
> > And for the network parameters, what is your optimizer and hyperparameters , including learning rate schedule (for CIFAR-10)?

---

> > > ### Author Response · Authors · 2018-12-08
> > > **Re: replacement=True or False?**
> > >
> > > Apologize for the mistake. The correct one is setting "replacement=False". Beta2 is set to be the default value in Pytorch (i.e., 0.999). As for network parameters, we use SGD optimizer with Nesterov momentum 0.9 and cosine learning rate schedule.

---

### Author Response · Authors · 2018-11-20
**Paper revision: new methods and new experiment results**

Hi all,

We have uploaded a revision of our paper with the following new methods and stronger experiment results:

a) “Economical alternative to mobile farm”. In Appendix C, we introduce an accurate latency prediction model and remove the need of building an expensive mobile farm infrastructure [1] when learning specialized neural network architectures for mobile phone. We add new experiment results on the mobile setting, where our model achieves state-of-the-art top-1 accuracy on ImageNet under mobile latency constraints.

b) “Make latency differentiable”. In Appendix D, we present a *differentiable* approach to handle the non-differentiable objectives (i.e. latency in our case). Specifically,  we propose the latency regularization loss based on our proposed latency prediction model. By incorporating the predicted latency of the network into the loss function as a regularization term, we are able to directly optimize the trade-off between accuracy and latency. We also add new experiments on ImageNet to justify the effectiveness of the proposed latency regularization loss.

[1] Tan, Mingxing, et al. "Mnasnet: Platform-aware neural architecture search for mobile." arXiv preprint arXiv:1807.11626 (2018).

---

### Public Comment · ~Robin_Tibor_Schirrmeister1 · 2018-11-23
**Further questions, also regarding REINFORCE**

Thanks for answering the questions so far, I also have some further questions.

1. What was the search time on CIFAR-10 in GPU hours? For Proxyless-R and Proxyless-G?
2. Is Batch Normalization in training or evaluation mode when optimizing architecture parameters?
3. For REINFORCE, what do you use as optimization metric on validation set for architecture parameters on CIFAR-10? Normal loss, like cross entropy or actually misclassification rate?
4. For REINFORCE, do you use any kind of baselining? Do you use multiple architecture samples per update? For example, right now I sample 10 architectures for each validation data batch and also subtract the mean metric/reward/loss before I compute the gradients.

---

> ### Author Response · Authors · 2018-12-08
> **Re: Further questions**
>
> Thanks for your questions. Please see responses below.
>
> >>> “What was the search time on CIFAR-10 in GPU hours? For Proxyless-R and Proxyless-G?”
>
> The search time depends on the size of the backbone architectures (e.g., number of blocks). For example, when searching with 54 blocks, it takes around 4 days on a single GPU for both Proxyless-R and Proxyless-G. When searching with fewer blocks (e.g. 8 blocks), it takes less than 1 day.
>
> >>> “Is Batch Normalization in training or evaluation mode when optimizing architecture parameters?”
>
> The batch normalization is in the training mode.
>
> >>> “For REINFORCE, what do you use as optimization metric on validation set for architecture parameters on CIFAR-10? Normal loss, like cross entropy or actually misclassification rate?”
>
> We use the misclassification rate. Normal loss, like cross entropy, may also be a feasible optimization metric
>
> >>> “For REINFORCE, do you use any kind of baselining? Do you use multiple architecture samples per update?”
>
> The baseline is the moving average of previous mean metrics with a decay of 0.99. And we update every 8 samples.

---

> > ### Public Comment · ~Robin_Tibor_Schirrmeister1 · 2018-12-21
> > **Thanks!**
> >
> > Thanks for the answers! Congratulations on acceptance!

---

### Public Comment · ~Miao_Zhang1 · 2019-04-04
**is it possible to release search code?**

Dear authors,
I am working NAS also, and I am very interested in this paper. However, I found that you just release pretrained models and evaluation code, is it possible to release the main search code in the future?

Sincerely

---

> ### Public Comment · (anonymous) · 2019-06-06
> **+1, please release code**
>
> It would be really helpful if you could release the full code for this project. Since you define a new search space that's good for one-shot methods, it could become a new benchmark if it's easy to use your code to do further experiments in this space.

---

### Public Comment · (anonymous) · 2019-06-28
**How to choose paths while training in ProxylessNAS？**

When we train the architecture parameters, is in every layer we sample two paths to update or only a single layer’s two path’s parameters will be update?

 If only one layer’s parameters update, how to choose paths of the rest layers? The highest-weight one or a random one?

---

### Public Comment · (anonymous) · 2020-02-03
**Architecture search code for CIFAR10 dataset**

Hi,
Can you please release the architecture search code for the CIFAR10 dataset also?

Thanks

---

### Meta-Review · Area_Chair1 · 2018-12-13
**Good empirical results. Novelty is limited.**

**Confidence:** 4
**Recommendation:** Accept (Poster)

**Metareview:**

This paper integrates a bunch of existing approaches for neural architecture search, including OneShot/DARTS, BinaryConnect, REINFORCE, etc. Although the novelty of the paper may be limited, empirical performance seems impressive. The source code is not available. I think this is a borderline paper but maybe good enough for acceptance.